# Anti-Inflammatory Activity of β-thymosin Peptide Derived from Pacific Oyster (*Crassostrea gigas*) on NO and PGE_2_ Production by Down-Regulating NF-κB in LPS-Induced RAW264.7 Macrophage Cells

**DOI:** 10.3390/md17020129

**Published:** 2019-02-21

**Authors:** Dukhyun Hwang, Min-jae Kang, Mi Jeong Jo, Yong Bae Seo, Nam Gyu Park, Gun-Do Kim

**Affiliations:** 1Department of Microbiology, College of Natural Sciences, Pukyong National University, Busan 48513, Korea; amitie725@naver.com (D.H.); rkdalsrbmc@naver.com (M.-j.K.); miss5274@naver.com (M.J.J.); haehoo76@pknu.ac.kr (Y.B.S.); 2Department of Biotechnology, College of Fishery Sciences, Pukyong National University, Busan 48513, Korea; ngpark@pknu.ac.kr

**Keywords:** β-thymosin, cytokines, NF-κB, nitric oxide, Pacific oyster (*Crassostrea gigas*), prostaglandin E_2_, RAW264.7 cells

## Abstract

β-thymosin is known for having 43 amino acids, being water-soluble, having a light molecular weight and ubiquitous polypeptide. The biological activities of β-thymosin are diverse and include the promotion of wound healing, reduction of inflammation, differentiation of T cells and inhibition of apoptosis. Our previous studies showed that oyster β-thymosin originated from the mantle of the Pacific oyster, *Crassostrea gigas* and had antimicrobial activity. In this study, we investigated the anti-inflammatory effects of oyster β-thymosin in lipopolysaccharide (LPS)-induced RAW264.7 macrophage cells using human β-thymosin as a control. Oyster β-thymosin inhibited the nitric oxide (NO) production as much as human β-thymosin in LPS-induced RAW264.7 cells. It also showed that oyster β-thymosin suppressed the expression of prostaglandin E_2_ (PGE_2_), inducible nitric oxide synthase (iNOS) and cyclooxygenase-2 (COX-2). Moreover, oyster β-thymosin reduced inflammatory cytokines such as tumor necrosis factor-α (TNF-α), interleukin-1β (IL-1β) and interleukin-6 (IL-6). Oyster β-thymosin also suppressed the nuclear translocation of phosphorylated nuclear factor-κB (NF-κB) and degradation of inhibitory κB (IκB) in LPS-induced RAW264.7 cells. These results suggest that oyster β-thymosin, which is derived from the mantle of the Pacific oyster, has as much anti-inflammatory effects as human β-thymosin. Additionally, oyster β-thymosin suppressed NO production, PGE_2_ production and inflammatory cytokines expression via NF-κB in LPS-induced RAW264.7 cells.

## 1. Introduction

Inflammation is a defense system that removes deleterious stimuli or microbial infections. When the inflammatory responses are initiated, the damaged tissue is rapidly repaired by eliciting the appropriate signals [1]. Inflammation has relevance to numerous diseases, such as rheumatoid arthritis, chronic bronchitis, asthma, and cancer [2,3]. Macrophages are activated by T cell products, endocytosis, and cytokines, which are affected by several extracellular signals [4,5]. Lipopolysaccharide (LPS) is known as an endotoxin in the cell wall of Gram-negative bacteria. LPS action launches the proliferation of macrophages, and causes expression of inducible nitric oxide synthase (iNOS) and cyclooxygenase-2 (COX-2) [6]. RAW264.7 macrophages cell line is the most extensively studied in LPS action [5,7].

Nitric oxide (NO) controls various functions in mammalian cells. Identified as an effector in macrophage induced cytotoxicity, NO is synthesized by iNOS from L-arginine catalysis [8]. iNOS is regulated by stimulated-cytokines, and increasing iNOS expression induces transcriptional activation in macrophage cells [9]. COX-2 plays an important role in the conversion of arachidonic acid to prostaglandins (PGs), which control immune functions [6]. Pro-inflammatory cytokines such as tumor necrosis factor-α (TNF-α), interleukin-1β (IL-1β) and interleukin-6 (IL-6) act on up-regulation of inflammation and are produced by stimulated macrophages [10]. Inhibitory κB (IκBα) is a nuclear factor-κB (NF-κB) inhibitory protein and phosphorylated IκBα is degraded by ubiquitination. Thereby, NF-κB can translocate into the nucleus that induces the expression of pro-inflammatory cytokines [1]. 

β-thymosin was isolated from the calf thymus first, and comprised of a single polypeptide in the thymic hormone [11]. β-thymosin is a small ubiquitous polypeptide, containing 43 amino acids and localized in the cytoplasm and nucleus [12]. The biological effects of β-thymosin have been studied and the results show that it suppresses osteoclastic differentiation [11], promotes angiogenesis [13], regulates sepsis [14] and has anti-inflammatory effects [15]. In contrast to the β-thymosin in vertebrates, research on β-thymosin homologous in invertebrates has been investigated in very few studies [16,17]. In particular, β-thymosin in marine invertebrates has been rarely studied. 

Our previous study suggested that homologous β-thymosin derived from the mantle of the Pacific oyster (*Crassostrea gigas*) had antimicrobial activity [18]. Therefore in this study, we examined the anti-inflammatory activity of oyster β-thymosin derived from the Pacific oyster (*Crassostrea gigas*) in LPS-induced RAW264.7 cells. Oyster β-thymosin has been effective in suppressing inflammation as human β-thymosin. Oyster β-thymosin decreased the production of NO and PGE_2_, as well as the expression of iNOS and COX-2 in LPS-induced RAW264.7 cells. Inflammatory cytokines, such as TNF-α, IL-1β and IL-6 were also reduced by oyster β-thymosin in LPS-stimulated RAW264.7 cells. Inflammatory mediators were activated via the NF-κB pathway, but oyster β-thymosin suppressed the expression of NF-κB in LPS-induced RAW264.7 cells. Therefore, this study suggests that oyster β-thymosin effectively inhibits inflammatory effects via NF-κB in LPS-induced RAW264.7 cells.

## 2. Results

### 2.1. Sequence Alignment of Oyster and Human β-thymosin

The N-terminal sequence alignment of oyster and human β-thymosin is shown in Figure 1A. Oyster β-thymosin sequence was presented in GenBank, no.KM924551 [18] and human β-thymosin was referred from NCBI NM_021109.3. It shows the similarity between oyster and human β-thymosin.

### 2.2. Effects of β-thymosin on Cell Viability

Before the study, the anti-inflammatory effects on RAW264.7 cells and the cytotoxicity of oyster and human β-thymosin was examined on the human keratinocyte HaCaT cells by MTT (3-(4,5-Dimethylthiazol-2-yl)-2,5-Diphenyltetrazolium Bromide) assay. Human β-thymosin was used as a control to compare with the effects of oyster β-thymosin. As shown in Figure 1B, oyster and human β-thymosin did not affect cell viability at 20 μM on both RAW264.7 and HaCaT cells.

### 2.3. Inhibition of NO, PGE_2_, iNOS and COX-2 Expression by β-thymosin on LPS-Induced RAW264.7 Cells

The oyster and human β-thymosin have no cytotoxicity, and we examined the effects of both β-thymosins for NO production in LPS-induced murine macrophage RAW264.7 cells. NO synthesis is dependent on iNOS expression, which is produced by cytokines or other stimuli. NO can affect to damage normal tissue cells which causes a pathogenic effect and involves several diseases [19]. Figure 2A shows that oyster and human β-thymosin decreased NO production in a dose dependent manner. Several studies have already been reported which state that the human β-thymosin influences inhibition of NO production [12,15]. Oyster β-thymosin is significantly effective to decrease NO production at 20 μM, as much as human β-thymosin, in a dose dependent manner. Therefore, the oyster β-thymosin used in this study had a similar effect to inhibit NO production like human β-thymosin. 

PGE_2_ was measured by an ELISA kit. PGs are important lipid mediators in inflammation and are synthesized by PG G/H synthase and COX enzyme. In a normal state, the expression of COX-2 is not detected, but with the stimulation of LPS or cytokines, COX-2 is revealed [20]. Figure 2B shows that production of PGE_2_ was decreased by the oyster β-thymosin in a dose dependent manner. At 20 μM, oyster β-thymosin suppressed the production of PGE_2_ more than human β-thymosin. Our results suggested that oyster β-thymosin affected inhibition of NO and PGE_2_ expressions.

We further investigated whether iNOS and COX-2 expression were decreased or not by Western blot analysis. Figure 2C,D shows that oyster β-thymosin highly inhibited iNOS and COX-2 expression. Oyster β-thymosin significantly decreased the expression of iNOS and COX-2 at 20 μM in LPS-induced RAW264.7 cells. Likewise, human β-thymosin also suppressed the expression of iNOS and COX-2 at 20 μM in LPS-stimulated RAW264.7 cells. Human β-thymosin inhibited iNOS expression clearly, however, it had little effect to decrease COX-2 expression. These results indicated that oyster β-thymosin suppressed NO production via iNOS expression, as well as PGE_2_ expression via COX-2 expression in a dose dependent manner. It implied that oyster β-thymosin had the effect of suppressing inflammatory mediator expression, which is as similarly effective as human β-thymosin.

### 2.4. Inhibition of Cytokines Production by Oyster β-Thymosin on LPS-Stimulated RAW264.7 Cells

As oyster β-thymosin decreases NO, PGE_2_, iNOS and COX-2 expression as much as human β-thymosin, we studied the effect of oyster β-thymosin on the expression of pro-inflammatory cytokines. Pro-inflammatory cytokines, such as TNF-α, IL-1β and IL-6 are involved in the up-regulation of inflammatory responses in macrophage cells [10]. The expression of TNF-α, IL-1β and IL-6 cytokines were measured by ELISA kits. Figure 3A–C show that oyster β-thymosin suppressed cytokines expression in a dose dependent manner. At 20 μM, the expressions of TNF-α, IL-1β and IL-6 cytokines were significantly decreased. Therefore, we confirmed that oyster β-thymosin inhibited the expression of TNF-α, IL-1β and IL-6 cytokines by Western blot analysis. As shown in Figure 3D, oyster β-thymosin inhibited TNF-α, IL-1β and IL-6 cytokines expression at 20 μM in LPS-stimulated RAW264.7 cells. These results explain that oyster β-thymosin decreased the expression of TNF-α, IL-1β and IL-6.

### 2.5. Inhibitory Effects of NF-κB Pathway by Oyster β-thymosin on LPS-stimulated RAW264.7 Cells

The transcription factor NF-κB plays an important role in the inflammatory response’s regulation such as cytokine expression, apoptosis regulation and cell proliferation [8]. NF-κB activates the transcription of iNOS, COX-2 and pro-inflammatory cytokines [21]. We investigated whether oyster β-thymosin suppressed the activation of NF-κB or not, by measuring its phosphorylated forms in the LPS-induced RAW264.7 cells using Western blot analysis. Without stimulation, NF-κB is in an inactive form by binding with the inhibitory protein of the IκB family in the cytoplasm. When the stimulation is triggered, IκB proteins are phosphorylated then NF-κB is translocated into the nucleus [22]. As shown in Figure 4A, oyster β-thymosin considerably decreased phosphorylation of IκBα as well as NF-κB in a dose dependent manner. We also examined the translocation of NF-κB from cytosol to the nucleus using immunofluorescent staining. Compared to the LPS-induced RAW264.7 cells, NF-κB translocation was suppressed by oyster β-thymosin treated cells (Figure 4B). These results showed that oyster β-thymosin suppressed phosphorylation of both NF-κB and IκBα, and prevented the translocation of NF-κB into the nucleus.

## 3. Discussion

β-thymosin has diverse biological activities such as suppression of osteoclastic differentiation [11], promotion of angiogenesis [13] and reduction of inflammation [15] in mammals. β-thymosin has already been studied for its anti-inflammatory effects to decrease several inflammatory mediators such as chemokines and macrophage inflammatory protein 2 (MIP-2). However, β-thymosin homologue in invertebrates is investigated in few studies [16,17]. The Pacific oyster (*Crassostrea gigas*) was analyzed that it has immune-related gene [23]. For example, it is examined defensin from the oyster (*Crassostrea gigas*), it is related in defense system against microbial infections [24]. In the previous study, we identified and purified the Pacific oyster (*Crassostrea gigas*) peptide, which is homologue β-thymosin and has anti-microbial activity [18]. In this present study, we examined the anti-inflammatory effect of oyster β-thymosin, compared to human β-thymosin, in LPS-induced RAW264.7 macrophage cells.

In most of the experimental animal models to study human diseases, overexpression of NO implicates pathological symptoms including diabetes, allograft rejection and endotoxemia [25] as well as pro-inflammatory effects that cause edema, cytotoxicity and vasodilation. iNOS is only expressed by stimuli, such as LPS, TNF-α or IL-1. NO is produced from L-arginine oxidation via NOS. Once the iNOS is expressed, it induces the production of a significant amount of NO [26]. iNOS inhibitors have been used as therapeutic agents to cure gastrointestinal diseases and arthritis, therefore, it is important to reduce iNOS expression to treat inflammatory related diseases [8,27]. Based on the results, oyster β-thymosin is effective in suppressing the production of NO at 20 μM (Figure 2A). This concentration has the same effect on human β-thymosin. It has already been studied that human β-thymosin influences the suppression of NO production [16], and our results suggest that oyster β-thymosin has an identical effect as human β-thymosin, although a few amino acids mismatched. NO production was inhibited via suppression of iNOS expression, as shown in Figure 2C,D. Oyster β-thymosin suppressed the iNOS expression in LPS-induced RAW264.7 cells in a dose dependent manner. Human β-thymosin also decreased iNOS expression in LPS-induced RAW264.7 macrophage cells. These findings suggest that oyster β-thymosin has the effect to decrease NO production via iNOS, which is an equivalent effect in human β-thymosin. It may be used as a source of a therapeutic agent for inhibiting NO production.

To screen anti-inflammatory effect, most studies examine the expression of iNOS as well as COX-2. Both key factors are highly related and important to resolve the inflammatory mechanism. COX-2 enzyme is induced by stimuli and converts arachidonic acid to PGs, which are significant lipid mediators to promote high levels of inflamed tissues [28,29,30]. Increased PGE_2_ levels have been reported for HIV, aging, cystic fibrosis or cancer in murine models and patients. COX-2 inhibitors could help to avoid overproduction of PGE_2_ in atherosclerosis, arthritis and fever [31]. Aspirin has been widely used as a nonsteroidal anti-inflammatory drug that selectively inhibits COX-2 [28], and it is important to find anti-inflammatory treatments which are able to regulate COX-2 expression. Our research showed that oyster β-thymosin suppressed PGE_2_ production, with a better effect than human β-thymosin (Figure 2B). The results were related to Figure 2C,D. It showed that oyster β-thymosin abolished COX-2 expression clearly, however, human β-thymosin had little effect to decrease COX-2 expression in a dose dependent manner. Compared to Figure 2C,D, oyster β-thymosin was much more effective in decreasing expression of inflammatory mediators, such as iNOS and COX-2. The results suggest oyster β-thymosin is effective in suppressing the production of PGE_2_ and COX-2 expression.

TNF-α, IL-1β and IL-6 are included in pro-inflammatory cytokines which are involved to mediate endogenous pyrogens, regulate inflammatory reaction and stimulate acute phase reactants [10]. Macrophage cells are stimulated by pro-inflammatory cytokines, then produce NO and PGs. Those cytokines could stimulate iNOS and COX-2 expression as well [26]. Pro-inflammatory cytokines are related to several diseases. For example, increased TNF-α expression is associated with the deterioration of renal function in uremia and IL-6 is related to the promotion of muscle wasting and atherosclerosis [32]. Those pro-inflammatory cytokines bind to specific receptors then activate transcription factors such as NF-κB, which play an important role in inflammation development [10]. Therefore, to mediate the expression of pro-inflammatory cytokines, we suggest a decisive treatment to ameliorate inflammatory related disease. Our research indicated that oyster β-thymosin significantly inhibited TNF-α, IL-1β and IL-6 cytokines expression in LPS-induced RAW 264.7 cells (Figure 3). Based on these results, oyster β-thymosin suppressed pro-inflammatory cytokines expression.

NF-κB modulate the expression of inflammation, innate and adaptive immunity, response of stress and progression of cancer [22]. In normal conditions, NF-κB is combined with IκBα, which exists in an inactive form, and is inhibited from translocating into the nucleus by IκBα. Under stimulation conditions such as an LPS trigger, phosphorylated IκBα activates NF-κB translocation into the nucleus and IκBα is then degraded [5,33]. NF-κB is a pivotal regulator of pro-inflammatory gene expressions such as cytokines, chemokines, iNOS and COX-2 and is involved in inflammatory diseases. In inflammatory airway diseases, for example, increased NF-κB was examined in airway epithelial cells as well as nuclear translocation, and it correlated with the expression of pro-inflammatory cytokines, iNOS and COX-2. Those circumstances are also observed in rheumatoid arthritis, helicobacter pylori-associated gastritis and inflammatory bowel disease as well. Therefore, the regulation of NF-κB expression is an effective way to treat diverse inflammatory diseases [21]. We determined that oyster β-thymosin suppressed LPS-induced phosphorylation of NF-κB and IκBα, in addition to the prevention of NF-κB nuclear translocation (Figure 4). In accordance with the regulation of oyster β-thymosin on NF-κB, oyster β-thymosin negatively controlled the expression of inflammatory mediators such as iNOS, COX-2 and cytokines in LPS-induced macrophage cells via NF-κB inactivation.

## 4. Materials and Methods 

### 4.1. Reagents

Oyster β-thymosin was synthesized based on the previous study [18]. Human β-thymosin and LPS (*Escherichia coli*) were obtained from Sigma-Aldrich (St. Louis, MO, USA). Fetal bovine serum (FBS) and Dulbecco’s modified eagle medium (DMEM) were purchased from Cellgro Mediatech, Inc. (Manassas, VA, USA). PGE_2_, TNF-α, IL-1β and IL-6 enzyme-linked immunosorbent assay (ELISA) kits were purchased from R&D Systems Inc. (Minneapolis, MN, USA). Most of the antibodies were obtained from Cell Signaling Technology Inc. (Danvers, MA, USA): iNOS (D6B6S) Rabbit mAb #13120, COX-2 (D5H5) XP® Rabbit mAb #12282, TNF-α Antibody #3707, NF-κB pathway: NF-κB Pathway Sampler Kit #9936 (except p-NF-κB), β-Actin (D6A8) Rabbit mAb #8457, anti-rabbit IgG, horseradish peroxidase (HRP)-linked Antibody #7074, anti-mouse IgG, HRP-linked Antibody #7076, anti-rat IgG, HRP-linked Antibody #7077 (Beverly, MA, USA) and p-NF-κB(p-NFκB p65 (27.Ser 536): sc-136548), IL-1β(IL-1β (B122): sc-12742), and IL-6(IL-6 (10E5): sc-57315) antibodies were purchased from Santa Cruz Biotechnology, Inc. (Santa Cruz, CA, USA).

### 4.2. Cell Culture

RAW264.7 (murine macrophage cells) and HaCaT (human keratinocyte cells) were obtained from the American Tissue Culture Collection (Manassas, VA, USA). Those cells were cultured in DMEM supplemented with 10% FBS, and 1% penicillin-streptomycin (PAA Laboratories GmbH, PA, Austria) with 5% CO_2_ at 37 ℃.

### 4.3. Cell Viability Assay

Cell viability assay was performed as previously described [34]. In brief, RAW264.7 cells and HaCaT cells were treated with either oyster or human β-thymosin at mentioned concentrations for 24 h. The 100 μL of medium was exchanged and 10 μL WST-1^®^ (Daeil Lab service, Seoul, Korea) was added into each well. After 3 h, the optical density (OD) was measured with the ELISA reader at 460 nm.

### 4.4. NO Assay

NO assay was performed as previously described [34]. To explain briefly, RAW264.7 cells were pretreated with either oyster or human β-thymosin at above-mentioned concentrations for 2 h and treated with LPS (1 μg/mL) for 24 h. To measure the NO in the culture medium, 100 μL of Griess reagent (Sigma-Aldrich, St. Louis, MO, USA) was added to 100 μL of the culture medium and incubated at room temperature for 10 min. The OD at 540 nm was examined with the ELISA reader (Molecular Devices, Silicon Valley, CA, USA).

### 4.5. PGE_2_ and Cytokine Production

The cells were pretreated with oyster β-thymosin for 2 h, then treated with LPS (1 μg/mL) for 24 h. The culture medium was harvested, then PGE_2_, TNF-α, IL-1β and IL-6 levels were analyzed by the ELISA assay kit according to the manufacturer’s instructions.

### 4.6. Western Blot Analysis

Western blot analysis was followed as previously described [34]. In brief, after incubation, RAW264.7 cells were collected and lysed with an ice-cold lysis buffer. Aliquots from each sample were separated by 12% SDS-polyacrylamide gel electrophoresis and transferred to a nitrocellulose membrane (PALL Life Sciences, Pensacola, MI, USA). The membrane was blocked with 5% skim milk in phosphate buffered saline Tween-20 (PBST) buffer for 1 h. After blocking, the membrane was incubated with each of the primary antibodies, which were diluted 1:1000 with 5% BSA in 1× PBST at 4 °C overnight. The blots were washed three times with PBST butter, and the membrane was incubated with HRP-conjugated second antibodies (anti-rabbit IgG, anti-mouse IgG or anti-rat IgG, diluted 1:2000 with 5% skim milk in 1× PBST) at room temperature for 1 h. The blots were washed three times using PBST butter and detected by an ECL solution^®^ (AbFontier, Gyeonggi, Korea).

### 4.7. Immunofluorescent Staining

Immunofluorescent staining was performed as previously described [34]. In brief, RAW264.7 cells were treated with 10 μM of oyster β-thymosin for 2 h and induced with LPS for 24 h on bottom dishes. The cells were stained with DAPI solution at 37 °C for 30 min and then fixed with 4% formaldehyde at room temperature for 15 min. Samples were blocked for 1 h in 5% rabbit normal serum (Santa Cruz Biotechnology, Inc., Santa Cruz, CA, USA). Cells were incubated with 0.1 μg/mL of NF-κB p65 antibody for 2 h in the dark, then incubated with 0.1 μg/mL of Alexa Fluor^®^ 488 Conjugate (Cell Signaling Technology Inc., Danvers, MA, USA) for another 1 h. The stained cells were mounted on the slides with Prolong Gold Antifade^®^ reagent (Invitrogen, Grand Island, NY, USA) and observed in a confocal laser scanning microscope (CarlZeiss LSM 700, Jena, Germany). 

### 4.8. Statistical Analysis

The analysis of sequence alignment was performed by MultAlin (http://multalin.toulouse.inra.fr/multalin/multalin.html), then the results were performed by ESPript 3.0 (http://espript.ibcp.fr) [35]. The results were expressed as mean ± standard error of the mean and performed in triplicates. Data were analyzed by ANOVA post hoc analysis and followed Dunnett’s multiple comparison tests. GraphPad Prism 6.0 (GraphPad, San Diego, CA, USA) was used for statistical analysis, *p* < 0.05 was considered statistically significant.

## 5. Conclusions

In our previous study, we demonstrated that oyster β-thymosin derived from the Pacific oyster (*Crassostrea gigas*) had antimicrobial activity [18] and here we investigated the anti-inflammatory effects of oyster β-thymosin compared to human β-thymosin in LPS-induced RAW264.7 macrophage cells. Oyster β-thymosin suppressed the protein expression levels of inflammatory mediators such as iNOS and COX-2, following inhibitions of NO and PGE_2_ production. The results were equivalent to human β-thymosin. Also, oyster β-thymosin decreased the expression of TNF-α, IL-1β and IL-6. Moreover, oyster β-thymosin inhibited the phosphorylation of both NF-κB and IκBα and translocation of NF-κB into the nucleus. These findings suggest that the Pacific oyster β-thymosin could be a potential candidate for an effective anti-inflammatory therapeutic agent.

## Figures and Tables

**Figure 1 marinedrugs-17-00129-f001:**
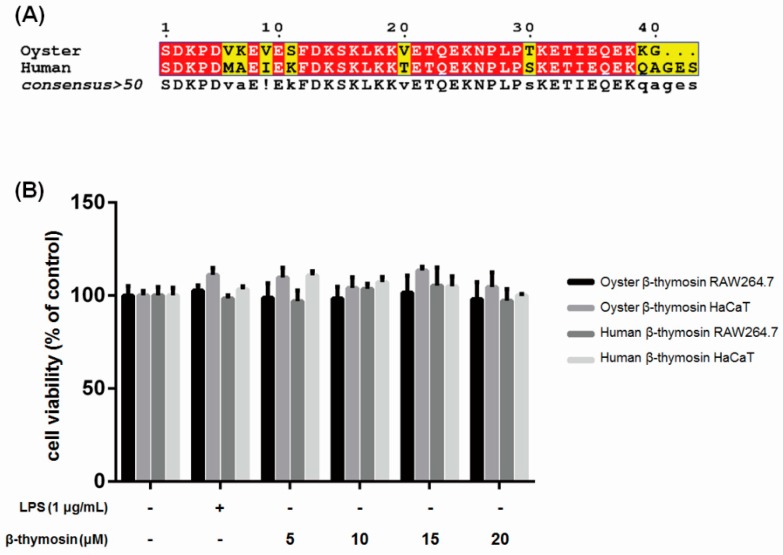
Sequence alignment of oyster β-thymosin and human β-thymosin and the effects of β-thymosin on the cell viability of RAW264.7 and HaCaT cells. (**A**) Sequences were analyzed by Multalin then performed by ESPript 3.0 (http://espript.ibcp.fr). The similar residues are written with black bold characters and boxed in yellow (! is either isoleucine or valine). (**B**) RAW264.7 and HaCaT cells were stimulated with LPS (1 μg/mL) for oyster and human β-thymosin for 24 h. The cell viability assay was performed in triplicates by WST-1^®^ (Daeil Lab Service, Gyeonggi, Korea) and the results are presented as the mean ± standard error.

**Figure 2 marinedrugs-17-00129-f002:**
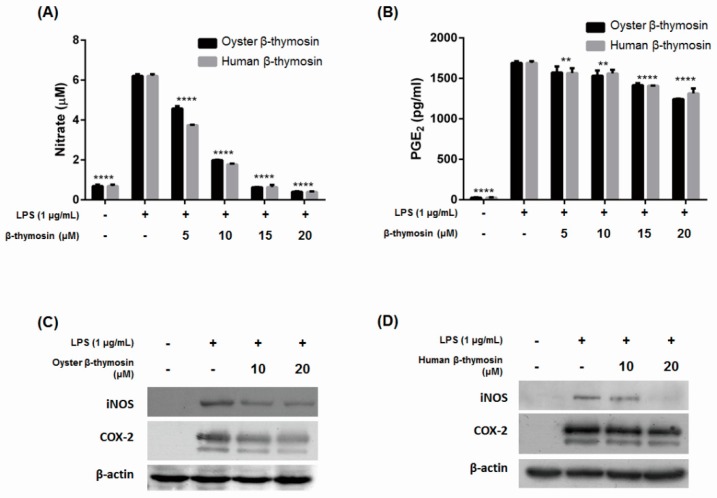
Effects of oyster and human β-thymosin on NO, PGE_2_ production and iNOS and COX-2 expression on LPS-induced RAW264.7 cells. Cells were pretreated with the designated concentration of each β-thymosin for 2 h and induced with LPS (1 μg/mL) for 24 h. (**A**) NO production was determined by Griess reagents; (**B**) PGE_2_ production was measured by the PGE_2_ ELISA kit; (**C**,**D**) was determined by Western blot analysis. β-actin was used as an internal control. The data represent the mean ± standard error of the mean of triplicate experiments. ** *p* < 0.01, **** *p* < 0.0001 vs. the LPS-induced group.

**Figure 3 marinedrugs-17-00129-f003:**
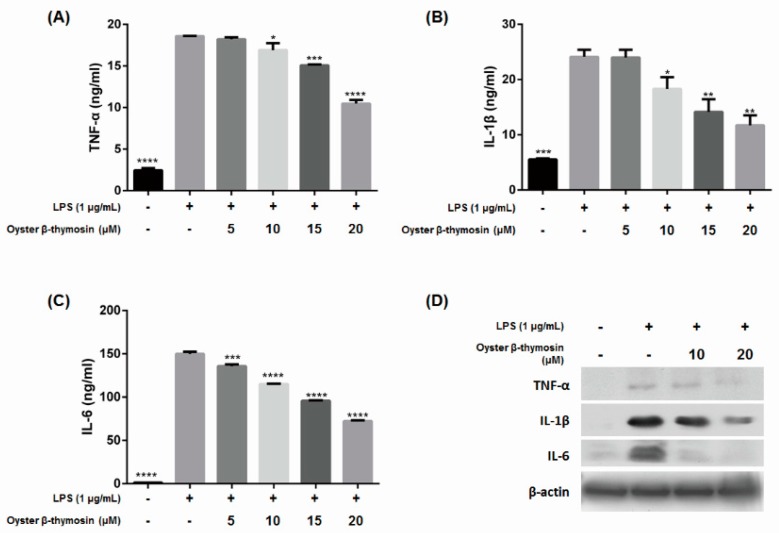
Effects of oyster β-thymosin on pro-inflammatory cytokines production in LPS-induced RAW264.7 cells. Cells were pretreated with oyster β-thymosin for 2 h then induced with LPS (1 μg/mL) for 24 h. After the stimulation by LPS, pro-inflammatory cytokines were released into the culture medium. The culture medium was collected and followed by analysis of TNF-α (**A**), IL-1β (**B**), and IL-6 (**C**) production by ELISA. (**D**) Western blot analysis using antibodies against TNF-α, IL-1β and IL-6, and β-actin was used as an internal control. The data represent the mean ± standard error of the mean of three independent experiments. * *p* < 0.05, ** *p* < 0.01, *** *p* < 0.001, **** *p* < 0.0001 vs. the LPS-induced group.

**Figure 4 marinedrugs-17-00129-f004:**
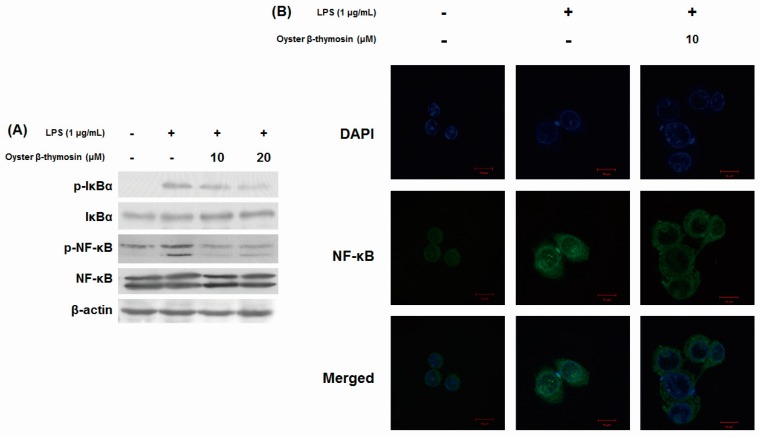
Effects of oyster β-thymosin on phosphorylation of either NF-κB or IκBα, and NF-κB translocation in LPS-stimulated RAW264.7 cells. Cells were pretreated with the designated concentration of oyster β-thymosin for 2 h and induced with LPS (1 μg/mL) for 2 h (**A**) and for 24 h (**B**). (**A**) Cell lysates were subjected to Western blot analysis using antibodies against p-IκBα, IκBα, p-NF-κB and NF-κB. β-actin was used as an internal control. (**B**) Translocation of NF-κB was detected by immunofluorescence staining. Cells were stained with DAPI (4′,6-diamidino-2-phenylindole) for visualization of nuclei (blue) and NF-κB immunofluorescence antibody (green), magnification × 1000, scale bar 10 µm.

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
