# Peer review of "Anti-Inflammatory Activity of β-thymosin Peptide Derived from Pacific Oyster (Crassostrea gigas) on NO and PGE2 Production by Down-Regulating NF-κB in LPS-Induced RAW264.7 Macrophage Cells"

_marinedrugs, 2019, doi:10.3390/md17020129_

Round 1
Reviewer 1 Report
Authors Hwang et al. in their MS are comparing effects of beta -thymosin peptides isolated either from Pacific oyster or a synthetic one on their potential to affect activation of mouse RAW264.7 macrophages by LPS. Authors report that beta -thymosin peptides have significant potential to inhibit selected markers of mouse macrophage activation including inducible nitric oxide synthase expression and nitric oxide production, TNF-alpha, IL-1beta, IL-6, or activation of NF-kappaB. Overall, the study is designed well and experiments are performed and presented correctly. However, data are not surprising considering previous studies clearly showing beta-thymosin anti-inflammatory properties and presence of thymosins in almost all phyla of the animal kingdom. Thus the study in current form is missing a significant novelty.
Major comments:
- Authors should clarify which aminoacid residues are responsible for the interaction with beta-thymosin receptors (by highlighting these AA in the figure 1). Are these AA conserved comparing human beta -thymosin peptide sequence and oyster beta -thymosin peptide sequence?
- Since mouse macrophages can differ from cells of human origin in their capability to recognize beta-thymosin of oyster origin authors should perform at least some experiments with human monocytes/macrophages to prove the presented concept.
- Are the effects of oyster originated beta -thymosin based on interaction with cell membrane receptors or are dependent on intracellular sequestration? How the extra- or intra-cellular concentrations declared to be effective under presented experimental conditions are comparable with endogenously produced thymosins in mouse macrophages?
- In vivo experiments that would prove the in vitro observations clarifying the potential of beta-thymosin of oyster origin to promote wound regeneration will significantly increase the novelty of the study.
Minor comments:
- The beginning of the introduction is confusing stating that inflammation is positive for would healing and then directly switch to negative sides of inflammation. This part of the introduction can be improved.
- Results - row 99: The meaning of the statement about L-arginine in the context of this study is unclear and can be omitted or better explained.
- The pictures documenting the immunofluorescence staining in Figure 4. should be provided brighter.
- The thorough check of the English (particularly the Discussion part) and typing should be performed (e.g. row 92: “cells” twice; row 173: in mammals; row 173: missing “for”; …)
- The extremely high significance of statistical tests (ANOVA combined with Dunnett’s multiple comparison test) in most figures with low number of repeats (n=3 mostly) is surprising. Authors should check for the correct application of the test and the depicted obtained significance.
Author Response
Major comments:
- Authors should clarify which aminoacid residues are responsible for the interaction with beta-thymosin receptors (by highlighting these AA in the figure 1). Are these AA conserved comparing human beta -thymosin peptide sequence and oyster beta -thymosin peptide sequence?
The information of human beta-thymosin peptide and oyster beta-thymosin is described by previous study. The previous study is “Nam, B. H.; Seo, J. K.; Lee, M. J.; Kim, Y. O.; Kim, D. G.; An, C. M.; Park, N. G. Functional analysis of Pacific oyster (Crassostrea gigas) β-thymosin: focus on antimicrobial activity. Fish Shellfish Immunol. 2015, 45, 167-174”.
- Since mouse macrophages can differ from cells of human origin in their capability to recognize beta-thymosin of oyster origin authors should perform at least some experiments with human monocytes/macrophages to prove the presented concept.
There are many studies which did experiment to use mouse macrophages. It may have some different capability of mouse and human macrophages. But if the capability was too much different between mouse and human macrophages, many researchers must use both mouse and human macrophages. If we focus on the difference between oyster and human beta-thymosin, we should compare in as many types of cells, but we would like to show oyster beta- thymosin has an anti-inflammatory effects as much as human beta-thymosin.
- Are the effects of oyster originated beta -thymosin based on interaction with cell membrane receptors or are dependent on intracellular sequestration? How the extra- or intra-cellular concentrations declared to be effective under presented experimental conditions are comparable with endogenously produced thymosins in mouse macrophages?
TLR4 (toll-like receptor 4) recognizes pathogen-associated molecular patterns and damage-associated molecular patterns. When the pathogen was invaded, TLR4 recognizes molecular patterns, which start to triggers to express the inflammation-related pathways (especially NF-κB pathways). Oyster beta-thymosin had an effect to decrease the expression of NF-κB pathways on LPS-induced RAW264.7 cells. So it may base on the interaction with cell membrane receptors.
It did not show the endogenously produced thymosins in mouse macrophages, however we showed the results of human beta-thymosin. The reference study (Sosne, G. et al. Thymosin beta 4 suppression of corneal NFκB: a potential anti-inflammatory pathway. Exp. Eye Res. 2007, 84, 663-669) compared the human beta-thymosin group and human beta-thymosin on TNF-α induced group. In this study, it described the human beta-thymosin suppressed NFκB pathways.
Other reference (Lee, S. I., et al. Thymosin beta-4 suppresses osteoclastic differentiation and inflammatory responses in human periodontal ligament cells. PloS one, 2016, 11, e0146708) used human beta-thymosin and showed the anti-inflammatory effect in vitro model of H2O2-stimulated PDLCs. In those studies, they showed the effect of human beta-thymosin effects. Our study, we used oyster beta-thymosin and it compared with human beta-thymosin. Oyster beta-thymosin had an anti-inflammatory effects as much as human beta-thymosin, so it may present the effect of oyster beta-thymosin on LPS-induced RAW264.7 cells.
- In vivo experiments that would prove the in vitro observations clarifying the potential of beta-thymosin of oyster origin to promote wound regeneration will significantly increase the novelty of the study.
Previous study, we researched the oyster beta-thymosin peptide. In this study, we would focus on the anti-inflammatory effect of oyster beta-thymosin via NF-κB signaling on RAW264.7 cells in vitro. In this study, the amount of oyster beta-thymosin was not much to do in vivo test. We are now preparing to mess up the oyster beta-thymosin to check the anti-inflammatory and other biological effect of oyster beta-thymosin in vivo study, therefore in this research, we focus on anti-inflammatory effect of oyster beta-thymosin via NF-κB on LPS-induced macrophage cells.
Minor comments:
- The beginning of the introduction is confusing stating that inflammation is positive for would healing and then directly switch to negative sides of inflammation. This part of the introduction can be improved.
Changed.
- Results - row 99: The meaning of the statement about L-arginine in the context of this study is unclear and can be omitted or better explained.
Changed.
- The pictures documenting the immunofluorescence staining in Figure 4. should be provided brighter.
- The thorough check of the English (particularly the Discussion part) and typing should be performed (e.g. row 92: “cells” twice; row 173: in mammals; row 173: missing “for”; …)
Changed.
- The extremely high significance of statistical tests (ANOVA combined with Dunnett’s multiple comparison test) in most figures with low number of repeats (n=3 mostly) is surprising. Authors should check for the correct application of the test and the depicted obtained significance.
We would show the difference to compare to LPS and other groups. The reason why we did Dunnett’s multiple comparison test. And our study was used as SEM, not to use SD, so changed correctly.
The revised version of manuscript was attached.

Reviewer 2 Report
Present work requires attention to following points:
Standard Abbreviations of Journal titles should be used. This should be uniform throughout the manuscript. e.g. Mediators Inflammation.
Some of the abbreviations in the manuscript require their full forms. e.g. DAPI
Reference section requires inclusion of latest literature up to 2016-2018.
e.g. Pardon MC. Anti-inflammatory potential of thymosin β4 in the central nervous system: implications for progressive neurodegenerative diseases. Expert Opin Biol Ther. 2018;18:165-169.
Ref9, title of journal is abbreviated wrong.
Ref27, title of journal requires correction. Also title seems to be incomplete. e.g. missing ‘Nitric oxide’.
Ref31 is missing article/page number.
Author Response
Standard Abbreviations of Journal titles should be used. This should be uniform throughout the manuscript. e.g. Mediators Inflammation.
Changed.
Some of the abbreviations in the manuscript require their full forms. e.g. DAPI
Changed.
Reference section requires inclusion of latest literature up to 2016-2018.
e.g. Pardon MC. Anti-inflammatory potential of thymosin β4 in the central nervous system: implications for progressive neurodegenerative diseases. Expert Opin Biol Ther. 2018;18:165-169.
Some changed.
Ref9, title of journal is abbreviated wrong.
Changed.
Ref27, title of journal requires correction. Also title seems to be incomplete. e.g. missing ‘Nitric oxide’.
Changed.
Ref31 is missing article/page number.
Changed.
The revised version of manuscript was attached file.

Reviewer 3 Report
The present manuscript deals with the effect on b-thymosin on murine macrophage cell model. The manuscript is easily readable but not very well written. The manuscript has to be carefully read and I highly recommend to let the manuscript correct with a professional English proofreader. To increase the quality of the manuscript, I have these comments that need to be fully addressed in the present paper:
- The cytotoxicity was determined just by MTT test that evaluates the metabolic activity. It is not clear if the tested compound would effect other commonly used cytotoxic endpoints. I highly recommend to measure more than just one cytotoxic endpoint.
- Line 85 – “Human β-thymosin was used as a control” – what control do the authors mean? According to Figs in the article, the negative control is just pure culture and LPS is a positive control. In addition, it is not obvious if the control contained the vehicle (solvent), the some that was used to dissolve b-Thymosin and LPS. Thus is is not clear if the used control is an appropriate control. Please address these questions in the article in detail.
- The replicates are technical or biological? The replicates should be always from different passages. Please comment.
- Line 105 – “PGE2 was also measured by ELISA kit”. The previous paragraph describes the measurement of NO using Griess reagent. So the word “ also” is not appropriate here.
- Line 118 – According to Figs 2A,B there is not much difference between the inflammatory regulators. Why do authors think that oyster b-thymosin is MUCH effective than human one?
- Line 119 – please correct “human oyster” b-thymosin.
- The authors must describe what antibody was used to detect e.g. cytokines. If the antibody used in WB is the same as in case of ELISA, than it does not bring additional prove. Please describe the WB and ELISA methodology in detail.
- I recommend to use LPS-stimulated rather than LPS-induced
- Line 213 and line 227 – These is not enough evidence that the tested substance could be good or bad potential medicine or alternative therapeutic agent.
- The methodology part is extremely week. According the present description, it would be impossible to repeat the experiment. Please rewrite the methodology and add more details to every point including experimental design, solvent, details in WB, dilution of medium, sample handling…..
Author Response
The cytotoxicity was determined just by MTT test that evaluates the metabolic activity. It is not clear if the tested compound would effect other commonly used cytotoxic endpoints. I highly recommend to measure more than just one cytotoxic endpoint.
MTT assay was shown the cytotoxicity effect. Whether oyster and human thymosin had cytotoxicity effect, it may need cytotoxic endpoints, however, there was no cytotoxicity effect both macrophage cells and normal cells. So it may not to need it.
- Line 85 – “Human β-thymosin was used as a control” – what control do the authors mean? According to Figs in the article, the negative control is just pure culture and LPS is a positive control. In addition, it is not obvious if the control contained the vehicle (solvent), the some that was used to dissolve b-Thymosin and LPS. Thus is is not clear if the used control is an appropriate control. Please address these questions in the article in detail.
Changed.
- The replicates are technical or biological? The replicates should be always from different passages. Please comment.
We did biological replicates. We purchased three cell stocks by one cell line from ATCC. After receiving the stocks, split out each stocks. Then did independent experiments to use different passages and different stock vials.
- Line 105 – “PGE2 was also measured by ELISA kit”. The previous paragraph describes the measurement of NO using Griess reagent. So the word “ also” is not appropriate here.
Changed.
- Line 118 – According to Figs 2A,B there is not much difference between the inflammatory regulators. Why do authors think that oyster b-thymosin is MUCH effective than human one?
Changed.
- Line 119 – please correct “human oyster” b-thymosin.
Changed.
- The authors must describe what antibody was used to detect e.g. cytokines. If the antibody used in WB is the same as in case of ELISA, than it does not bring additional prove. Please describe the WB and ELISA methodology in detail.
Changed.
- I recommend to use LPS-stimulated rather than LPS-induced
Some changed. But both are same meaning.
- Line 213 and line 227 – These is not enough evidence that the tested substance could be good or bad potential medicine or alternative therapeutic agent.
Changed.
- The methodology part is extremely week. According the present description, it would be impossible to repeat the experiment. Please rewrite the methodology and add more details to every point including experimental design, solvent, details in WB, dilution of medium, sample handling…..
There was reference so it showed briefly. But rewrote and added details.
The revised version of manuscript was attached.

Round 2
Reviewer 1 Report
Authors answered positively to formal questions only. The comments and suggestions directed to improve novelty and clarify a mechanism how can oyster beta-thymosin affects macrophages were left unanswered. Unfortunately, these responses are confirming the limited novelty and originality of this work.
Based on their responses:
Part of information was already published in their previous paper “Nam, B. H.; Seo, J. K.; Lee, M. J.; Kim, Y. O.; Kim, D. G.; An, C. M.; Park, N. G. Functional analysis of Pacific oyster (Crassostrea gigas) β-thymosin: focus on antimicrobial activity. Fish Shellfish Immunol. 2015, 45, 167-174”.
Further, as authors highlighted in their response the conclusions of this study are limited to artificial mouse leukemia cell line RAW264.7. Macrophages of other species including humans not necessarily have to respond to oyster beta-thymosin.
They do not provide information about receptors for beta-thymosine. Generally, authors do not provide information about mechanism how beta-thymosine is recognized by macrophages.
Author Response
Authors answered positively to formal questions only. The comments and suggestions directed to improve novelty and clarify a mechanism how can oyster beta-thymosin affects macrophages were left unanswered. Unfortunately, these responses are confirming the limited novelty and originality of this work.
Based on their responses:
Part of information was already published in their previous paper “Nam, B. H.; Seo, J. K.; Lee, M. J.; Kim, Y. O.; Kim, D. G.; An, C. M.; Park, N. G. Functional analysis of Pacific oyster (Crassostrea gigas) β-thymosin: focus on antimicrobial activity. Fish Shellfish Immunol. 2015, 45, 167-174”.
Further, as authors highlighted in their response the conclusions of this study are limited to artificial mouse leukemia cell line RAW264.7. Macrophages of other species including humans not necessarily have to respond to oyster beta-thymosin.
Thank you for your comment. In the previous paper, it only showed analysis of oyster beta-thymosin and their antimicrobial activity. In this paper, we concentrate to use RAW264.7 cells and show anti-inflammatory activities of oyster beta-thymosin on LPS-induced RAW264.7 cells. Oyster beta-thymosin had anti-inflammatory effects as much as human beta-thymosin on LPS-induced RAW264.7 cells.
They do not provide information about receptors for beta-thymosin.
In this paper, we would show the anti-inflammatory activity of oyster beta-thymosin, not receptor interaction. We would like to focus on this study as oyster beta-thymosin had the effect to decrease the expression of inflammatory mediators such as iNOS, COX-2, NO, cytokines via NF-κB pathways on LPS-induced RAW264.7 cells.
Generally, authors do not provide information about mechanism how beta0thymosin is recognized by macrophages.
We studied anti-inflammatory effect of oyster beta-thymosin via NF-κB pathways on LPS-induced RAW264.7 cells. In the manuscript, we mentioned the mechanism in results part (2.5) and discussion part. To briefly explain in here, NF-κB is combined with IκBα in the normal state, so it exists in inactive form. When NF-κB is stimulated by such as LPS, iNOS, COX-2 and pro-inflammatory cytokines are expressed. Our results showed that oyster beta-thymosin inhibited the iNOS, COX-2, pro-inflammatory cytokines and NF-κB pathways expression on LPS-induced RAW264.7 cells.
The attached file is revised version of manuscript.

Reviewer 3 Report
Dear Authors,
thank you for your effort to correct your manuscript. Some of my comments were not understand correctly or/and not adrresed by you.
- there is always need to combine several cytotoxic tests to show that the sample is non-toxic. Each test just determine one specific toxic endpoint. Combination of cytotoxic tests can prove that you work in non-toxic range.
- line 81 - the corrected sentence does not make sense
- There is no comment in your response that you significantly improve English in your manuscript. It is really necessary to let the english-speaking person to fully correct your manuscript.
- please add the details of the used antibody as I asked in my previous review (company, catalog number...)
- please, if you correct your manuscript, write also details what was changed to the answer for reviewer.
Author Response
- There is always need to combine several cytotoxic tests to show that the sample is non-toxic. Each test just determine one specific toxic endpoint. Combination of cytotoxic tests can prove that you work in non-toxic range.
Thank you for your comment. Like as last comment, MTT assay is shown the cytotoxicity effect. Our oyster and human thymosin did not show cytotoxicity effect by the study concentration (20 μM). It is not mentioned in the manuscript, but for the first time, we did MTT test (0 ~ 50 μM of the oyster thymosin) on RAW264.7 cells, it did not showed any cytotoxicity activity on RAW264.7 cells. Then we investigated the experimental concentrations.
We did MTT test, however, we would like to cover the limitation of MTT test on RAW264.7 cells, so we use HaCaT normal cells. If oyster or human thymosin peptide showed the cytotoxicity effect on HaCaT cells, we could not progress for further studies. Both oyster and human thymosin showed no cytotoxicity effect both RAW264.7 macrophage cells and HaCaT normal cells so we could do further studies.
As you mentioned the comment, doing the combination of cytotoxic test is possible to prove our work. However, we did MTT assay both on RAW264.7 cells and HaCaT cells, and it would show there is no cytotoxicity. In this reference, which is similar study as ours, Ha, T. M., et al. (2017). Anti-Inflammatory Effects of Curvularin-Type Metabolites from a Marine-Derived Fungal Strain Penicillium sp. SF-5859 in Lipopolysaccharide-Induced RAW264. 7 Macrophages. Marine drugs, 15(9), 282., they used RAW264.7 cells only and did MTT test then showed the anti-inflammatory effects of their test peptide. Our study used both RAW264.7 and HaCaT cells and did MTT assay, so it would show no toxicity and cover the limitation of one cytotoxic test.
- line 81 – the corrected sentence does not make sense
Changed;
Before study the anti-inflammatory effects on RAW264.7 cells, the cytotoxicity of oyster and human β-thymosin was examined on HaCaT cells by MTT assay.
- There is no comment in your response that you significantly improve English in your manuscript. It is really necessary to let the english-speaking person to fully correct your manuscript.
We got the English-editing from native person.
- Please add the details of the used antibody as I asked in my previous review (company, catalog number…)
There is written the company and dilution information already. The details of antibody information is here.
Cell signaling Technology Inc.
- iNOS: iNOS (D6B6S) Rabbit mAb #13120
- COX-2: Cox2 (D5H5) XP® Rabbit mAb #12282
- TNF-α: TNF-α Antibody #3707
- NF-κB pathway: NF-κB Pathway Sampler Kit #9936 (except p-NF-κB)
- β-Actin: β-Actin (D6A8) Rabbit mAb #8457
- Anti-rabbit IgG, HRP-linked Antibody #7074
- Anti-mouse IgG, HRP-linked Antibody #7076
- Anti-rat IgG, HRP-linked Antibody #7077
Santa Cruz Biotechnology Inc.
- p- NF-κB: p-NFκB p65 (27.Ser 536): sc-136548
- IL-1β: IL-1β (B122): sc-12742
- IL-6: IL-6 (10E5): sc-57315
Changed;
Oyster β-thymosin was synthesized based on the previous study [18]. Human β-thymosin and LPS (Escherichia coli) were obtained from Sigma-Aldrich (St. Louis, MO, USA). Fetal bovine serum (FBS) and Dulbecco’s modified eagle medium (DMEM) were purchased from Cellgro Mediatech, Inc. (Manassas, VA, USA). PGE2, TNF-α, IL-1β, and IL-6 enzyme-linked immunosorbent assay (ELISA) kits were purchased from R&D Systems Inc. (Minneapolis, MN, USA). Most of the antibodies were obtained from Cell Signaling Technology Inc.: iNOS (D6B6S) Rabbit mAb #13120, COX-2 (D5H5) XP® Rabbit mAb #12282, TNF-α Antibody #3707, NF-κB pathway: NF-κB Pathway Sampler Kit #9936 (except p-NF-κB), β-Actin (D6A8) Rabbit mAb #8457, anti-rabbit IgG, HRP-linked Antibody #7074, anti-mouse IgG, HRP-linked Antibody #7076, anti-rat IgG, HRP-linked Antibody #7077 (Beverly, MA, USA) and p-NF-κB(p-NFκB p65 (27.Ser 536): sc-136548), IL-1β(IL-1β (B122): sc-12742), and IL-6(IL-6 (10E5): sc-57315) antibodies were purchased from Santa Cruz Biotechnology, Inc. (Santa Cruz, CA, USA).
- Please, if you correct your manuscript, write also details what was changed to the answer for reviewer.
It is mentioned or used red color.
The attached file is revised version of manuscript.

Round 3
Reviewer 3 Report
Dear authors,
I have no further comments.